# Inhibitory Effects of Gliadin Hydrolysates on BACE1 Expression and APP Processing to Prevent Aβ Aggregation

**DOI:** 10.3390/ijms252313212

**Published:** 2024-12-09

**Authors:** Chin-Yu Lin, Cheng-Hong Hsieh, Pei-Yu Lai, Ching-Wei Huang, Yung-Hui Chung, Shang-Ming Huang, Kuo-Chiang Hsu

**Affiliations:** 1Department of Biomedical Sciences and Engineering, Tzu Chi University, No. 701, Sec. 3, Zhongyang Rd., Hualien City 970374, Taiwan; geant@mail.tcu.edu.tw; 2Department of Nutrition, China Medical University, No. 100, Sec. 1, Jingmao Rd., Beitun Dist., Taichung City 40604, Taiwan; hyweuan@asia.edu.tw (C.-H.H.); azazmj2323@gmail.com (P.-Y.L.); u112211002@cmu.edu.tw (C.-W.H.); carol510020@gmail.com (Y.-H.C.); 3Department of Food Nutrition and Health Biotechnology, Asia University, 500 Lioufeng Rd., Wufeng, Taichung City 41354, Taiwan

**Keywords:** Alzheimer’s disease, amyloid beta, BACE1, gliadin, peptide

## Abstract

Alzheimer’s disease (AD), a leading neurodegenerative disorder, is closely associated with the accumulation of amyloid-beta (Aβ) peptides in the brain. The enzyme β-secretase (BACE1), pivotal in Aβ production, represents a promising therapeutic target for AD. While bioactive peptides derived from food protein hydrolysates have neuroprotective properties, their inhibitory effects on BACE1 remain largely unexplored. In this study, we evaluated the inhibitory potential of protein hydrolysates from gliadin, whey, and casein proteins prepared using bromelain, papain, and thermolysin. Through in vitro and cellular assays, bromelain-hydrolyzed gliadin (G-Bro) emerged as the most potent BACE1 inhibitor, with an IC_50_ of 0.408 mg/mL. G-Bro significantly reduced BACE1 expression and amyloid precursor protein (APP) processing in N2a/PS/APP cell cultures, suggesting its potential to attenuate Aβ aggregation. The unique peptide profile of G-Bro likely contributes to its inhibitory effect, with proline residues disrupting β-sheets, lysine residues introducing positive charges that hinder aggregation, hydrophobic residues stabilizing binding interactions, and glutamine residues enhancing solubility and stability. These findings highlight gliadin hydrolysates, particularly G-Bro, as potential natural BACE1 inhibitors with applications in dietary interventions for AD prevention. However, further studies are warranted to elucidate specific peptide interactions and their bioactivity in neural pathways to better understand their therapeutic potential.

## 1. Introduction

The increase in life expectancy among the elderly has corresponded with a rise in chronic diseases, particularly Alzheimer’s disease (AD), which has become a significant global health concern [1]. AD, a neurodegenerative disorder characterized by progressive cognitive decline, memory loss, and functional impairment, is the most common cause of dementia among older adults, affecting millions worldwide. As the global population ages, the prevalence of AD is expected to increase exponentially, underscoring the urgent need for effective preventive and therapeutic strategies [2,3]. However, progress in developing therapies that could potentially cure or prevent the onset of the disease remains limited [4].

A hallmark of AD is the accumulation of amyloid-beta (Aβ) plaques in the brain, which is believed to contribute to the clinical symptoms experienced by patients [5,6,7]. Targeting the rate-limiting enzyme responsible for Aβ production, β-secretase (BACE1), has emerged as a promising therapeutic strategy [8]. Unfortunately, many BACE1 inhibitors developed over the past decade have failed to reach clinical application, primarily due to undesirable side effects associated with their use such as cognitive impairments, peripheral neuropathy, and liver toxicity [9]. It has been suggested that amyloid deposition may begin years before the manifestation of apparent clinical symptoms, indicating the necessity for early intervention and preventive treatment strategies [10]. These challenges highlight the need for innovative approaches to BACE1 inhibition that can maintain long-term enzyme suppression while minimizing adverse effects. Consequently, recent research has turned toward naturally derived compounds, which are being evaluated for their potential to safely inhibit BACE1 over prolonged periods [11,12,13]. Such compounds may offer a multi-targeted approach that addresses the multifactorial nature of AD and aligns with the need for safer, more effective therapies to delay or prevent disease progression [11].

Recent advancements in protein research have highlighted the potential of bioactive peptides derived from protein hydrolysates as promising candidates for AD therapeutics. These peptides, which are generated through the hydrolysis of various protein sources, have been shown to possess neuroprotective activities that may counteract the pathological processes associated with AD [14]. Evidence suggests that bioactive peptides can regulate protein–protein interactions and modulate several biochemical pathways relevant to the disease’s progression [15]. Protein hydrolysates obtained from diverse sources, including marine proteins [16], plant proteins [17], and meat proteins [18], have demonstrated neuroprotective effects by reducing oxidative stress, enhancing antioxidant enzyme activity, and decreasing apoptosis in both in vitro and in vivo models. Furthermore, specific studies have identified the efficacy of protein hydrolysates in targeting pathways involved in AD pathogenesis, indicating their potential for preventative and therapeutic applications [19]. Recent findings also suggest that collagen and fibroin hydrolysates can improve cognitive functions, such as memory and language, in healthy adults [20].

This study aimed to investigate the inhibitory effects of food-derived protein hydrolysates from gliadin, whey, and casein on BACE1 activity and APP metabolism to prevent Aβ aggregation, a critical factor in AD progression. Protein hydrolysates were prepared using bromelain, papain, and thermolysin enzymatic digestion, and their effects were assessed for BACE1 inhibition in vitro and further evaluated in a N2a/PS/APP neuroblastoma cell model. By exploring the potential of naturally derived peptides to disrupt Aβ formation, this research firstly provides valuable insights into dietary-based preventive strategies for AD and highlights the therapeutic potential of food-derived protein hydrolysates.

## 2. Results and Discussion

### 2.1. Effects of Proteases on the Hydrolysis of Protein Substrates

The functional properties of protein hydrolysates are influenced significantly by the degree of hydrolysis (DH), underscoring the importance of precise DH monitoring to ensure optimal peptide distribution and functionality [21]. In this study, gliadin, whey, and casein proteins were hydrolyzed with bromelain, papain, and thermolysin, generating nine hydrolysate variations. The DH was determined using leucine as a standard, which allowed for reliable quantification of cleaved peptide bonds [22]. As shown in Figure 1, the hydrolysis curves displayed an initial high reaction rate attributed to the cleavage of surface-accessible polypeptide chains, followed by a gradual plateau. This trend aligns with findings from previous studies, where core proteins in compact forms were hydrolyzed at slower rates [23].

A progressive decline in available peptide bonds led to a plateau in DH, indicating saturation of the enzymatic reaction [24,25]. Toward the end of the hydrolytic curve, a secondary increase in DH was observed, possibly due to microbial activity or the conversion of insoluble protein components to soluble forms [26,27]. Since non-enzymatic factors may influence hydrolysates generated beyond this point, these later-stage samples were excluded as representative enzymatic hydrolysates for this study. Thus, the optimal hydrolysis durations were determined as the time points just before this secondary DH increase. As shown in Table 1, the optimal hydrolysis times were determined to be 2 h for papain, 3 h for bromelain, and 1 h for thermolysin.

The different DH values among enzyme–protein combinations reflect variations attributable to enzyme specificity, protein structure, and reaction conditions. A higher DH was observed with thermolysin, likely due to its broad specificity, which facilitates cleavage at peptide bonds with hydrophobic residues at the N-terminal [28]. Conversely, lower DH values observed with papain and bromelain may be attributed to reaction pH levels close to the isoelectric points of whey (pH 5.2) and casein (pH 4.6), potentially leading to protein aggregation that limits enzymatic accessibility [29,30]. Similarly, the low solubility of gliadin within the pH range of 6–8 further explains its comparatively low DH [31,32]. These findings underscore the necessity of optimizing hydrolysis conditions to maximize DH and enhance the functional properties of protein hydrolysates.

### 2.2. Inhibitory Activity of Protein Hydrolysates on BACE1

The BACE1 inhibitory potential of various protein hydrolysates was assessed through the BACE1 FRET assay. Table 2 shows that all hydrolysates derived from papain and bromelain exhibited inhibitory activity against BACE1 activity, except for thermolysin-derived hydrolysates (C-The, W-The, and C-The). The inhibition rate increased with higher concentrations (0.1–1 mg/mL). Moreover, at the same concentration level, the BACE1 inhibition rates of bromelain-derived hydrolysates (C-Bro, W-Bro, and C-Bro) were significantly higher than those of the papain-derived hydrolysates (C-Pap, W-Pap, and C-Pap). Further, Table 2 shows the IC_50_ values of the protein hydrolysates. The highest inhibitory rate was observed for G-Bro, with an IC_50_ value of 0.408 mg/mL, followed sequentially by W-Bro, G-Pap, W-Pap, and C-Bro. Hydrolysates such as C-The, W-The, and C-The demonstrated negligible inhibitory activity. Consequently, G-Bro was selected for further testing in subsequent cellular experiments.

Presently, several small-molecule BACE1 inhibitors have been identified, including AZD3293, E2609, CNP520, JNJ-54861911, and MK-8931. Despite their potent inhibition, adverse effects pose substantial limitations for clinical applications [33]. Protein hydrolysates, derived from natural sources, are posited as alternatives with potentially fewer side effects. Nonetheless, only limited studies have evaluated protein hydrolysates as BACE1 inhibitors. Notably, shrimp waste protein hydrolysates (IC_50_ = 0.54 mg/mL) and skate skin protein hydrolysates (IC_50_ = 0.56 mg/mL) have been shown to inhibit BACE1 activity [34,35]. In comparison, the IC_50_ of G-Bro (0.408 mg/mL) is marginally lower than these protein hydrolysates. Although the inhibitory activities of individual peptides were not evaluated in this study, it is anticipated that peptide purification could yield sequences with enhanced BACE1 inhibition.

Our results suggest that BACE1 inhibition by protein hydrolysates depends significantly on the type of protein and the enzyme used for hydrolysis. Hydrolysates generated by bromelain and papain displayed moderate to high inhibitory activity, whereas thermolysin-generated hydrolysates exhibited minimal inhibition. Gliadin-derived hydrolysates, particularly G-Bro, were identified as having the highest inhibitory activity, while casein-derived hydrolysates demonstrated relatively lower efficacy. The observed variations in inhibitory efficacy may arise from enzyme-specific cleavage patterns and differences in protein sequences, resulting in peptides with varied inhibitory potencies [36]. Research on BACE1 peptide inhibitors has indicated that no single amino acid residue universally contributes to inhibition, with peptide size and amino acid positioning playing crucial roles in potency [37]. These insights emphasize the importance of carefully selecting starting materials and hydrolysis enzymes to yield bioactive peptides with optimized inhibitory effects.

### 2.3. Effect of G-Bro on the Viability of N2a/PS/APP Cells

To assess the BACE1 inhibitory potential of G-Bro in a neuroblastoma cell model, a murine neuroblastoma cell line (N2a/PS/APP), which stably overexpresses the Swedish mutant form of human APP (APPswe, K595N/M596L) and PS2 mutation, was utilized. This cell line has been shown to express APPswe with increased affinity for BACE1, resulting in elevated levels of soluble APP (sAPP) and Aβ, making it a widely accepted model for investigating AD [38].

Before evaluating BACE1 inhibition, an MTT assay was conducted to determine the cytotoxicity of G-Bro in N2a/PS/APP cells across a range of concentrations (0.2 to 10 mg/mL) over 24, 48, and 72 h. Results indicated that G-Bro at concentrations between 0.5 and 5 mg/mL had minimal impact on cell viability, with only slight reductions observed as both concentration and exposure duration increased (Figure 2). In contrast, a significant reduction in cell viability was observed at the 10 mg/mL concentration after 72 h, with cell viability decreasing to 58.17% ± 7.3% (*p* < 0.01). Consequently, concentrations ranging from 0.5 to 2 mg/mL were determined to be appropriate for subsequent experimental applications.

### 2.4. Inhibitory Effect of G-Bro on BACE1 Protein Expression

The impact of G-Bro on BACE1 protein expression was assessed in N2a/PS/APP cells treated with concentrations of 0.2, 0.5, 1, and 2 mg/mL for 48 h. As shown in Figure 3, a significant dose-dependent reduction in BACE1 protein levels was observed, with the highest concentration of 2 mg/mL producing the most pronounced inhibition compared to untreated controls. Numerous natural compounds have demonstrated potential in attenuating BACE1 protein expression, which holds therapeutic promise for AD. Notably, studies on curcumin [39], luteolin [40,41], and icaritin [42] have reported substantial BACE1 suppression, highlighting the prospective utility of natural products in AD intervention. However, only one other study has examined the influence of protein hydrolysates on BACE1 protein expression. Zhang et al. (2019) demonstrated that protein hydrolysates derived from royal jelly significantly downregulated BACE1 expression in N2a/APP695 cells, likely via modulation of histone deacetylation [38]. Additional evidence suggests that protein hydrolysates may modulate protein expression by influencing upstream transcriptional factors [43,44].

### 2.5. Effects of G-Bro on sAPP Production and Aβ Aggregation

To explore the impact of G-Bro on downstream APP processing, extracellular levels of soluble APP (sAPP) were quantified using an ELISA assay. As shown in Figure 4, a dose-dependent reduction in extracellular sAPP secretion was observed, with G-Bro treatments of 1, 2, and 5 mg/mL downregulating sAPP production by 0.87-fold, 0.63-fold, and 0.55-fold, respectively, compared to untreated cells (*p* < 0.05).

Given that only aggregated Aβ species are associated with toxicity in AD, further investigation was conducted to assess the anti-aggregative effects of G-Bro on Aβ. As shown in Figure 5, G-Bro at 1, 2, and 5 mg/mL significantly inhibited Aβ aggregation by 0.78-fold, 0.76-fold, and 0.67-fold, respectively, compared to control (*p* < 0.05). Furthermore, G-Bro treatment shifted the balance between Aβ monomers and oligomers, promoting monomer stability while inhibiting oligomerization, thereby favoring the presence of non-toxic Aβ monomers.

### 2.6. Identification of G-Bro Peptides by nanoUHPLC-ESI-Q TOF Mass Spectrometry

The peptide sequences in G-Bro were analyzed using Q-TOF mass spectrometry to elucidate the potential mechanisms by which G-Bro exerts its anti-aggregation effects on Aβ. The acquired spectra were cross-referenced against the Swiss-Prot database to identify matching sequences. The identified peptides within G-Bro are summarized in Table 3, revealing seven peptides ranging from 9 to 39 amino acids in length.

Peptides have consistently demonstrated their ability to disrupt Aβ aggregate formation in various studies [45]. Specifically, prior research has highlighted the efficacy of protein hydrolysates, such as colostrinin—derived from bovine colostrum—and whey protein hydrolysates, in inhibiting the self-assembly of Aβ42 fibrils [46]. The mechanisms underlying this inhibitory effect have been attributed to proline-rich polypeptides in these hydrolysates [47].

The unique structural properties of proline allow it to function effectively as a β-sheet breaker. Being the only amino acid lacking a hydrogen atom on the α-amino group, proline is unable to participate in the hydrogen bonding that stabilizes the Aβ β-sheet structure. Moreover, the geometric configuration of peptide bonds within the β-sheet motifs of Aβ is incompatible with the torsion angles of the peptidyl–prolyl bond induced by proline [48]. The cyclic structure of proline further inhibits its integration into the hydrogen bonding network of the β-sheet, as it does not sterically fit within this arrangement [48,49]. Consequently, studies have indicated that proline residues are typically absent from the interior of β-sheet structures, suggesting that amyloidogenic sequences may be intolerant to proline [50].

The ability of the listed peptides to disrupt Aβ assembly is thought to be influenced by a range of mechanisms associated with their amino acid residues. Proline (P) residues, present in peptides such as SPQRPGQGQQPGQGQQGYYPTSPQQPGQWQQPEQGQPRY, GAAGEPGK, VRVPVPQLQPQNPSQQQPQK, and LVGALVLPSK, are particularly significant. It is well established that proline introduces kinks in peptide chains due to its unique cyclic structure, which hinders the formation of stable β-sheets typically associated with Aβ aggregation [48]. Consequently, the presence of proline is believed to obstruct the alignment of Aβ monomers into oligomers and fibrils, thereby reducing aggregate formation and potentially interfering with the nucleation of fibril growth.

In addition to proline, the presence of lysine (K) residues in several of these peptides, including GAAGEPGK, ENQILLK, and SRRYLLKK, is also critical. With its positive charge, lysine may engage in electrostatic interactions with the negatively charged regions of Aβ peptides, stabilizing the peptide structure in a way that could prevent aggregation. The influence of terminal lysines on the solubility and conformation of these peptides is considered significant in promoting their ability to inhibit Aβ assembly [51]. Furthermore, lysine residues may also be involved in forming non-covalent interactions, such as hydrogen bonds or ionic interactions, with surrounding amino acids, thereby affecting the overall stability and conformation of the peptide [52].

The presence of glutamine (Q) residues in the identified peptides may also play a role in their ability to disrupt Aβ assembly. Glutamine is known for its polar and hydrophilic nature, which can influence peptide solubility and interactions with surrounding molecules [53]. The abundance of glutamine in sequences such as SPQRPGQGQQPGQGQQGYYPTSPQQPGQWQQPEQGQPRY, GAAGEPGK, and VRVPVPQLQPQNPSQQQPQK can contribute to several vital mechanisms. Firstly, the hydrophilic nature of glutamine can enhance the solubility of the peptides in aqueous environments, helping to prevent the aggregation of Aβ by facilitating the formation of stable, non-aggregated conformations. The ability of glutamine to form hydrogen bonds with water and other polar molecules can also contribute to stabilizing peptide structures in solutions [54].

Also, multiple glutamine residues may promote intramolecular hydrogen bonding within the peptides. This could lead to unique secondary structures that further impede the aggregation of Aβ [55]. Such structural variations may disrupt the orderly packing of Aβ monomers into fibrils, as the conformational flexibility provided by the glutamine residues may inhibit the rigid structures typically observed in amyloid fibrils [56]. Additionally, glutamine’s ability to engage in specific interactions with Aβ could alter the conformational dynamics of the aggregates. By introducing steric hindrance and disrupting hydrogen bonding patterns essential for Aβ aggregation, glutamine residues may contribute to the overall inhibitory effect of these peptides on Aβ assembly [57].

Hydrophobic amino acids, such as lysine (L), valine (V), glycine (G), alanine (A), and leucine (L) found in LVGALVLPSK, are expected to facilitate interactions with the hydrophobic regions of Aβ, disrupting aggregation processes [58]. These hydrophobic interactions are believed to play a pivotal role in altering the conformational dynamics of Aβ, potentially shifting the equilibrium away from aggregation-prone conformations. In particular, hydrophobic residues may promote the formation of a less structured or more flexible peptide conformation, which could be less favorable for Aβ aggregation [59]. Incorporating hydrophilic and charged residues, such as serine (S) and glutamic acid (E), further enhances the peptides’ potential to solubilize Aβ aggregates and prevent the formation of larger, insoluble fibrils. These residues can increase the overall polarity of the peptide, thereby improving solubility in aqueous environments, which is crucial for the effective inhibition of Aβ aggregation [60].

Additionally, the sequence diversity among the identified peptides is expected to allow for various interactions with Aβ, resulting in distinct conformations that may disrupt the assembly of toxic forms. The unique combination of amino acid sequences in these peptides may facilitate the formation of specific secondary structures that do not support the aggregation process, thus contributing to their overall inhibitory effects on Aβ assembly [60].

In summary, the synergistic effects of the structural properties of proline (P), the positive charge of lysine (K) residues, the role of hydrophobic interactions, and the contributions of glutamine (Q) residues to solubility and structural stability are believed to contribute to the inhibition of Aβ assembly and aggregation collectively. This understanding may suggest potential pathways for therapeutic intervention in amyloid-related diseases, highlighting the importance of targeting specific peptide characteristics in the development of effective treatments. Further investigation into the specific interactions and conformational dynamics of these peptides, mainly focusing on the roles of proline and glutamine, is warranted to elucidate the precise mechanisms underlying their protective effects against aggregation.

## 3. Materials and Methods

### 3.1. Study Design

The research framework diagram can be found in Figure 6.

### 3.2. Materials

Gliadin protein was purchased from Sigma-Aldrich (St. Louis, MO, USA), while casein and whey protein isolates were obtained from MyProtein (Cheshire, UK). The enzymes used for hydrolysis, including bromelain (EC 3.4.22.32), papain (EC 3.4.22.2), and thermolysin (EC 3.4.24.27), were procured from Sigma-Aldrich. These enzymes were selected based on their distinct proteolytic properties, allowing for producing a range of peptides with varying sizes and sequences. Other reagents, such as the BACE1 fluorescence resonance energy transfer (FRET) assay kit, Dulbecco’s Modified Eagle Medium (DMEM), fetal bovine serum (FBS), antibiotics, and antibodies, were obtained from standard commercial suppliers.

### 3.3. Preparation of Protein Hydrolysates

Protein hydrolysates were prepared using gliadin, whey, and casein. For each hydrolysis, 1 g of the protein was dissolved in 12.5 mL of Milli-Q water and stirred until fully dissolved, ensuring no precipitation. The pH was adjusted to the optimal values for the respective enzymes using 2N NaOH or 2N HCl: bromelain (pH 6.7), papain (pH 6.2), and thermolysin (pH 8.0). The enzyme-to-substrate ratio was maintained at 3%. Before the addition of enzymes, the protein solutions were pre-incubated in a water bath for 15 min to reach the enzymes’ optimal temperature.

The hydrolysis reaction was conducted in a water bath shaker at 125 rpm for bromelain and papain and at 110 rpm for thermolysin. Samples were collected at predetermined time points based on the enzyme used: for bromelain, at 0, 1, 2, 3, 4, and 5 h; for papain, at 0, 0.5, 1, 1.5, 2, and 3 h; and for thermolysin, at 0, 20, 40, 60, 80, and 100 min. To stop the hydrolysis, 4 mL of the hydrolysate was heated to 100 °C for 15 min and then rapidly cooled on ice. The hydrolysates were centrifuged at 12,100× *g* for 10 min to separate the supernatant, which was either used for a degree of hydrolysis (DH) analysis or stored for further processing. Absorbance at 420 nm was measured using a spectrophotometer with L-leucine as the standard to analyze DH. The DH was then calculated using the following equation:$$DH\;(\%)=\frac{\left(L_{t}-L_{0}\right)}{\left(L_{max}-L_{0}\right)}\times 100$$
where:

*L_t_* = The amount of free amino acids at different time points.

*L*_0_ = The amount of free amino acids at 0 min of protein hydrolysis.

*L_max_* = Free amino acid content.

### 3.4. BACE1 Activity Assay

The inhibitory activity of gliadin hydrolysates against BACE1 was evaluated using a fluorescence resonance energy transfer (FRET)-based BACE1 assay. In this assay, the principle revolves around the cleavage of a fluorogenic substrate by the BACE1 enzyme, which releases a fluorescent signal measurable via a plate reader. The experimental setup involved mixing 20 μL of gliadin hydrolysate, dissolved in sodium acetate buffer, with 20 μL of BACE1 enzyme and 20 μL of a fluorogenic substrate in a 96-well black plate. The reaction mixture was then incubated at room temperature for 60 min and kept in the dark to prevent interference from ambient light. Fluorescence measurements were taken at two-time points: initially at 0 min and again at 60 min, using excitation and emission wavelengths of 530 nm and 590 nm, respectively. The percentage of BACE1 inhibition was calculated using the formula:$$Inhibition\;\left(\%\right)=\left[1-\frac{\left(S-S_{0}\right)}{\left(C-C_{0}\right)}\right]\times 100$$
where *S* and *S*_0_ represent the fluorescence readings of the sample at 60 and 0 min, while *C* and *C*_0_ correspond to the control readings at the same time points.

IC_50_ (mg/mL) represents the protein hydrolysate concentration required for 50% inhibition of BACE1 activity. SigmaPlot 14.0 was used to obtain a line of best fit and an equation for calculating the IC_50_ value.

### 3.5. Cell Viability

The N2a/PS/APP cell line [7], which carries presenilin 2 and human amyloid precursor protein (hAPP) Swedish K595N/M596L mutations, was cultured in Dulbecco’s Modified Eagle Medium—High Glucose (DMEM-H) supplemented with 10% fetal bovine serum (FBS) and 1% penicillin–streptomycin. Cultures were maintained in a humidified 37 °C incubator with 5% CO_2_, with cells passaged every 2–3 days upon reaching approximately 80% confluency. For viability assessments, cells were seeded in 12-well plates and allowed to adhere for 24 h before treatment. Cells were then exposed to various concentrations of protein hydrolysates for 24, 48, and 72 h. Following treatment, 600 μL of 0.5 mg/mL MTT reagent was added to each well and incubated for 2 h at 37 °C to facilitate formazan crystal formation. Subsequently, 600 μL of isopropanol was added to dissolve the formazan crystals, and the solution was gently mixed by tapping the plate. An aliquot of 100 μL was then transferred to a 96-well plate, and absorbance was measured at 570 nm using a microplate reader. Cell viability was calculated according to the formula:$$Percentage\;of\;viable\;cells\;\left(\%\right)=\left(\frac{{Abs}_{sample}}{{Abs}_{control}}\right)\times 100$$

### 3.6. β-Secretase (BACE1) Protein Expression

Cell lysates were collected and prepared for analysis by mixing with sample buffer and denaturing at 95 °C for 10 min. The samples were then cooled on ice for 10 min before being stored at −20 °C. Protein samples were separated on a 10% SDS-PAGE gel and transferred to polyvinylidene difluoride (PVDF) membranes. PVDF membranes were pre-activated with methanol prior to transfer.

The membranes were blocked with phosphate-buffered saline with Tween-20 (PBST) containing 5% skim milk for 1 h at room temperature for immunoblotting. Primary antibodies, including anti-BACE1 and anti-GAPDH as a loading control, were then applied to the membranes, which were incubated overnight at 4 °C on a rocking platform. Following incubation with primary antibodies, membranes were washed with PBST for 1 h at room temperature, then incubated with horseradish peroxidase (HRP)-conjugated secondary antibodies for an additional hour on a rocking platform. After a final wash with PBST for 1 h at room temperature, protein bands were visualized using a Western Lightning chemiluminescence reagent and detected with the ChemiDoc™ XRS+ Imaging System (Bio-Rad, Hercules, CA, USA). Band intensity was quantified using ImageJ Version 1.54 software.

### 3.7. The Levels of sAPP and Aβ

The levels of sAPP and aggregated Aβ in culture media were quantified using the DuoSet Human APP ELISA kit (R&D Systems in Waltham, MA, USA, Catalog No. DY850) and the Human Aggregated Beta-Amyloid ELISA kit (Invitrogen, Waltham, MA, USA). Media were collected after 24, 48, and 72 h of treatment, centrifuged at 500× *g* for 5 min at 4 °C to remove cellular debris, and stored at −80 °C for subsequent analysis.

For the sAPP assay, the capture antibody was diluted to 4 μg/mL and used to coat a 96-well plate with 100 μL per well, followed by overnight incubation at room temperature. After three washes with Wash Buffer (300 μL), wells were blocked with 300 μL of Reagent Diluent for 1 h at room temperature. Diluted samples (10-fold in Reagent Diluent) and standards were added in duplicate (100 μL/well) and incubated at room temperature for 2 h. After incubation and washing, 100 μL of the detection antibody (300 ng/mL) was added and incubated for 2 h. After washing, 100 μL of Streptavidin-HRP solution (diluted 200-fold in Reagent Diluent) was added to each well and incubated for 20 min in the dark. MB substrate solution (100 μL) was added to develop the reaction for 20 min, and Stop Solution (50 μL) was then used to terminate the reaction. Absorbance was read at 450 nm with correction at 540 nm to eliminate background noise.

For Aβ detection, 100 μL of standard or diluted samples were added to each well of the ELISA plate, followed by incubation for 2 h at room temperature. Wells were washed four times with Wash Buffer (300 μL) and then incubated with 50 μL of Hu Aggregated Aβ Biotin Conjugate Solution for 1 h. Following washes, 100 μL of Streptavidin-HRP solution (prepared by diluting 100× Anti-Rabbit IgG HRP in HRP diluent) was added and incubated for 30 min in the dark. The reaction was developed with 100 μL of Stabilized Chromogen and terminated with 100 μL of Stop Solution. Absorbance was measured at 450 nm.

### 3.8. Peptide Identification of Protein Hydrolysates

Peptide identification was performed using a maXis Impact™ quadrupole time-of-flight (Q-TOF) mass spectrometer (Bruker Daltonics, Bremen, Germany) equipped with electrospray ionization (ESI). This mass spectrometry system comprises three key components: an ionization source, a mass analyzer, and a detector. During analysis, analytes of interest are first ionized in the ionization source. These ionized fragments are subsequently analyzed by the mass analyzer, where they are exposed to an electric field that manipulates their trajectories according to their mass-to-charge ratio (*m*/*z*), allowing for separation and detection in the mass detector.

In tandem mass spectrometry (MS/MS), selected ions undergo an additional round of fragmentation, providing a second stage of mass analysis that offers enhanced structural information on the peptide fragments [61]. The generated data are processed to produce mass spectra, then matched against sequences in the Swiss-Prot database to identify peptide sequences.

### 3.9. Statistical Analysis

Data are expressed as mean ± standard deviation (SD). Statistical analysis was conducted using one-way analysis of variance (ANOVA). Significance thresholds were set at *p* < 0.05, indicated by *, and *p* < 0.01, indicated by **.

## 4. Conclusions

This study demonstrated the potential of protein hydrolysates, particularly G-Bro derived from gliadin, in modulating key biological processes linked to Alzheimer’s disease. Among tested hydrolysates, G-Bro showed superior BACE1 inhibitory activity and effectively reduced extracellular levels of soluble APP (sAPP) while preventing Aβ aggregation. The inhibitory effect of G-Bro on Aβ formation and aggregation may be attributed to a combination of its distinct peptide characteristics. Proline (P) residues likely disrupt β-sheet structures, lysine (K) residues introduce positive charges that interfere with aggregation, hydrophobic interactions stabilize peptide binding, and glutamine (Q) residues enhance solubility and structural stability. Together, these properties create a multifaceted mechanism that effectively hinders Aβ assembly, highlighting the potential of G-Bro as a therapeutic candidate in addressing amyloid pathology. Overall, these findings suggest that G-Bro holds significant promise for further investigation as a potential therapeutic candidate for Alzheimer’s disease.

While this study demonstrated the potential of G-Bro derived from gliadin in modulating key biological processes related to Alzheimer’s disease, several limitations exist. First, this study primarily focused on in vitro assays, and the results may not fully reflect the complexity of in vivo conditions. The therapeutic potential of G-Bro needs to be validated through animal models and clinical trials to assess its efficacy and safety in living organisms. Additionally, the precise molecular mechanisms underlying the observed effects, such as the role of individual peptide residues in modulating Aβ aggregation, require further investigation. This study also does not explore the potential long-term effects of G-Bro treatment, which are crucial for its development as a therapeutic candidate. Lastly, while G-Bro shows promise in inhibiting BACE1 and preventing Aβ aggregation, the interaction with other pathways involved in Alzheimer’s disease pathology, such as tau aggregation or neuroinflammation, was not investigated in this study. Future research should consider these factors to provide a more comprehensive understanding of G-Bro’s potential as a therapeutic agent for Alzheimer’s disease.

## Figures and Tables

**Figure 1 ijms-25-13212-f001:**
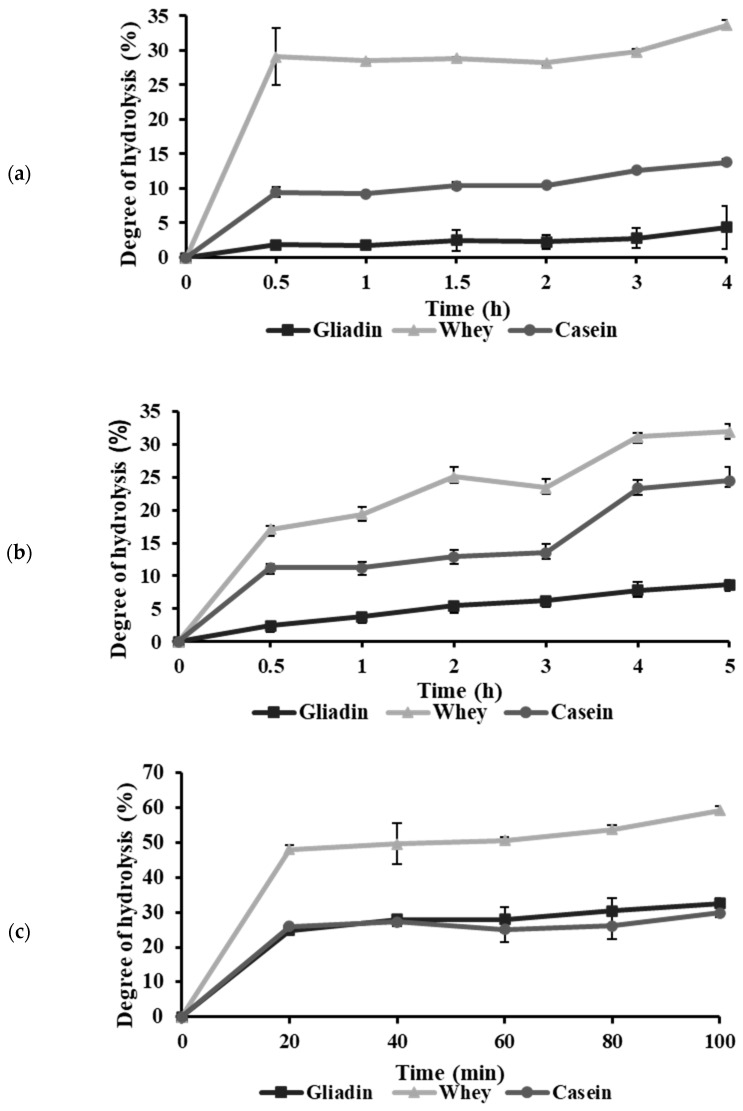
Degree of hydrolysis of proteins hydrolyzed by (**a**) papain, (**b**) bromelain, and (**c**) thermolysin.

**Figure 2 ijms-25-13212-f002:**
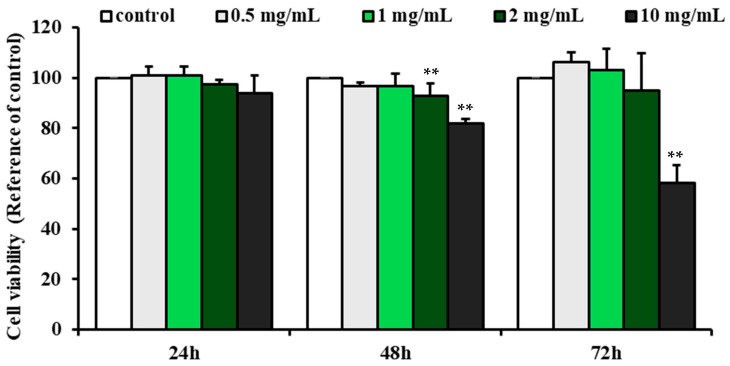
Effect of G-Bro on the viability of N2a/PS/APP cells. Data are expressed as mean ± SD of three independent experiments. Statistical analyses were performed using one-way ANOVA test. Non-treated cells were considered as control. ** *p* < 0.01 vs. control.

**Figure 3 ijms-25-13212-f003:**
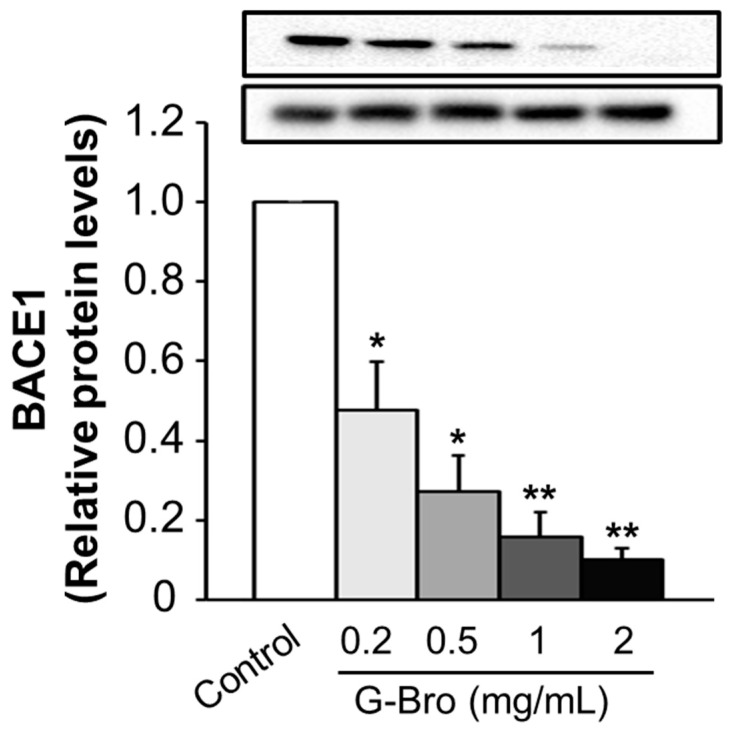
Effect of G-Bro on BACE1 expression in N2a/PS/APP cells. Data are expressed as mean ± SD of three independent experiments. Statistical analyses were performed using one-way ANOVA test. Band intensity was evaluated with Image J Version 1.54. Non-treated cells were considered as control. * *p* < 0.05 and ** *p* < 0.01 vs. control.

**Figure 4 ijms-25-13212-f004:**
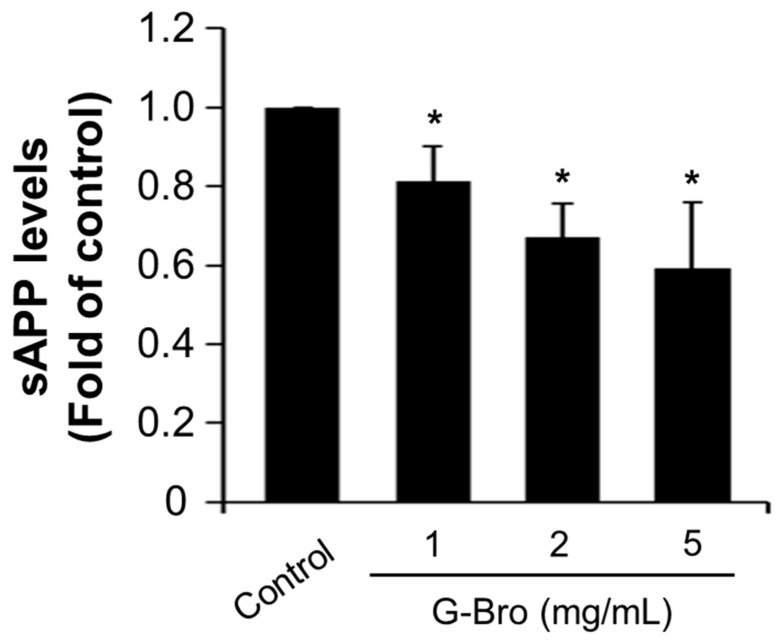
Effect of G-Bro on sAPP production in N2a/PS/APP cells. Data are expressed as mean ± SD of three independent experiments. Statistical analyses were performed using one-way ANOVA test. Non-treated cells were considered as control. * *p* < 0.05 vs. control.

**Figure 5 ijms-25-13212-f005:**
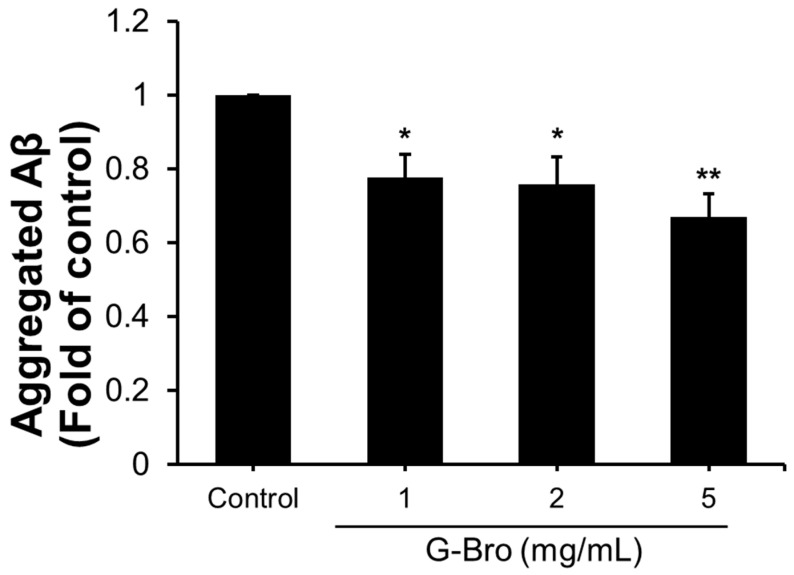
Effect of G-Bro on Aβ aggregate formation in N2a/PS/APP cells. Data are expressed as mean ± SD of three independent experiments. Statistical analyses were performed using one-way ANOVA test. Non-treated cells were considered as control. * *p* < 0.05 and ** *p* < 0.01 vs. control.

**Figure 6 ijms-25-13212-f006:**
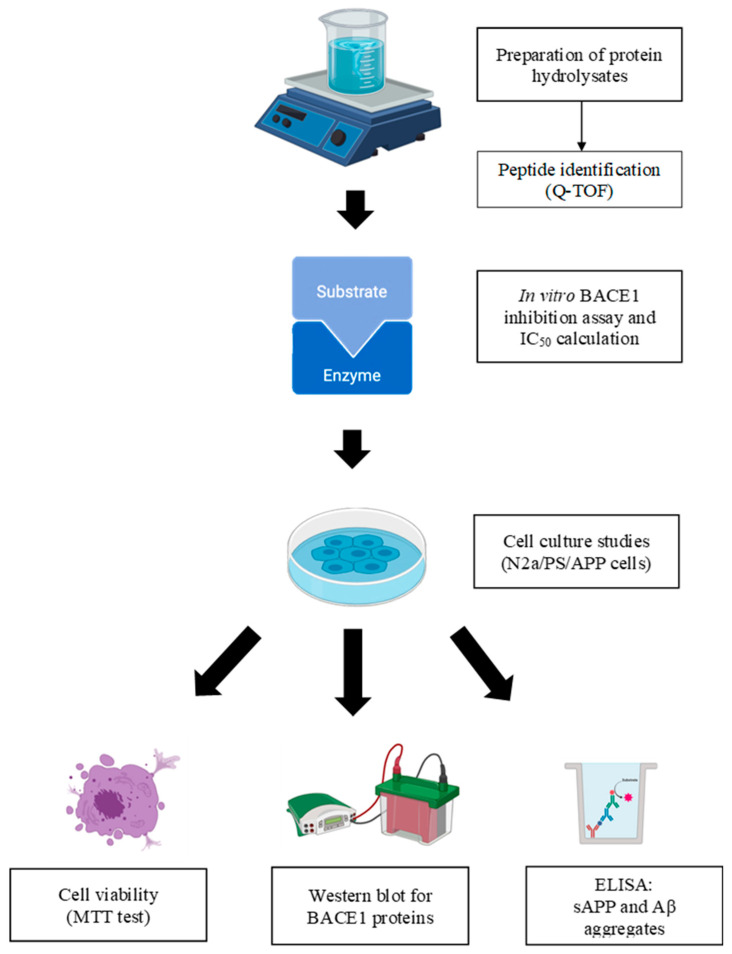
The overall framework of the study.

**Table 1 ijms-25-13212-t001:** Optimal hydrolysis duration for protein hydrolysates.

Abbreviation	Protein	Enzyme	Optimal Hydrolysis Duration	DH (%)
G-Pap	Gliadin	Papain	2 h	1.56
W-Pap	Whey	Papain	2 h	28.19
C-Pap	Casein	Papain	2 h	10.41
G-Bro	Gliadin	Bromelain	3 h	4.58
W-Bro	Whey	Bromelain	3 h	25.09
C-Bro	Casein	Bromelain	3 h	13.69
G-The	Gliadin	Thermolysin	1 h	28.27
W-The	Whey	Thermolysin	1 h	51.27
C-The	Casein	Thermolysin	1 h	26.41

**Table 2 ijms-25-13212-t002:** IC_50_ of protein hydrolysates for BACE1 inhibition.

Protein Hydrolysates	Equation	R^2^	IC_50_ (mg/mL)
G-Pap	y = 46.3x + 18.9	0.974	0.672
W-Pap	y = 51.5x + 14.3	0.996	0.693
C-Pap	y = 17.9In(x) + 42.3	0.929	NA
G-Bro	y = 54.6x + 27.7	0.931	0.408
W-Bro	y = 49.5x + 23.7	0.950	0.533
C-Bro	y = 37.9x + 15	0.971	0.923
G-The	y = −107x + 7.86	0.983	NA
W-The	y = −187x + 6.76	0.906	NA
C-The	y = −531x + 89.14	0.977	NA

NA: not applicable. Data presented as NA indicate that the inhibition of BACE1 was still below 50% even at maximum concentration of protein hydrolysate. BACE1 inhibitory rates of protein hydrolysates were measured at concentration of 0.1, 0.25, 0.5 0.75, and 1 mg/mL.

**Table 3 ijms-25-13212-t003:** Peptide fragments from G-Bro identified by Q-TOF analysis.

Peptide No	*m*/*z* Meas.	Rt [min]	Score	Range	Sequence
1	1095.5222	19.45	81.25	130–168	SPQRPGQGQQPGQGQQGYYPTSPQQPGQWQQPEQGQPRY
2	686.3484	15.36	27.47	426–433	GAAGEPGK
3	766.4068	21.22	59.75	21–40	VRVPVPQLQPQNPSQQQPQK
4	857.4509	43.51	21.15	534–540	ENQILLK
5	1063.6136	39.33	20.97	1615–1622	SRRYLLKK
6	879.4480	36.82	35.77	248–255	DIMGVSNK
7	996.6132	30.76	40.96	16–25	LVGALVLPSK

## Data Availability

The data presented in this study are available on request from the corresponding author.

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
