# Peer review of "Inhibitory Effects of Gliadin Hydrolysates on BACE1 Expression and APP Processing to Prevent Aβ Aggregation"

_ijms, 2024, doi:10.3390/ijms252313212_

Round 1
Reviewer 1 Report
Comments and Suggestions for Authors
To whom it may concern,
Please address the following issues before resubmitting the article for publication:
1. I suggest you prepare a structured abstract (background; materials and method; results; conclusion).
2. "Unfortunately, many BACE1 inhibitors developed over the past decade have failed to reach clinical application, primarily due to undesirable side effects associated with their use" - can you develop this in one more phrase? examples of medication and adverse effects?
3. In the materials and method section, I suggest you add first a "study design" subsection where you explain the "big picture" of your study.
4. Please add the ethical consent for this study (mandatory for publication) both in the "materials and method" section and at the end of the manuscript.
5. State clearly the limitations of your study!
6. Devide the conclusion section in two paragraphs: the first one, where you conclude the results of your work, and the second one, where you suggest future research directions.
Author Response
The point-by-point response to the reviewer’s comments has been uploaded in the attachment.

Reviewer 2 Report
Comments and Suggestions for Authors
The reviewer's comments have been attached

Author Response

(The authors gave the same response as above.)

Round 2
Reviewer 1 Report
Comments and Suggestions for Authors
The manuscript was improved according to my previous comments.
The decision for publication remains to the Editors.